# Brief communication: Depth-averaging of 3D depth-resolved MPM simulation results of geophysical flows for GIS visualization

Hervé Vicari[1,2,3], Michael Lukas Kyburz[1,2,3], and Johan Gaume[1,2,3]

[1]WSL Institute for Snow and Avalanche Research SLF, Davos Dorf, Switzerland
[2]Climate Change, Extremes, and Natural Hazards in Alpine Regions Research Center CERC, Davos Dorf, Switzerland
[3]Institute for Geotechnical Engineering, ETH Zürich, Zürich, Switzerland

**Correspondence:** Hervé Vicari (herve.vicari@slf.ch)

**Abstract.** Significant advances in full-3D modeling of geophysical flows have provided deeper insights into complex processes and predictive potential. However, practical application in the natural hazard community remains limited due to inadequate GIS integration of simulation results. This study addresses the oxymoronic transformation of 3D *depth-resolved* MPM simulation outputs into simplified *depth-averaged* results, such as flow depth and thickness, and slope-parallel and slope-normal velocities. Specifically, we present an algorithm that rasterizes scattered MPM outputs into a 2D format, enhancing their utility for hazard mapping and mitigation. We demonstrate our approach by applying it to an ice avalanche event, which is simulated using MPM and visualized in GIS. Notably, the 3D MPM shows slope-normal flow velocity components over terrain jumps, and our algorithm enables the identification of flow detachment from the terrain, which depth-averaged models typically neglect.

## 1 Introduction

With the emergence of computers, numerical modelling of geophysical flows (e.g., debris flows, rock, snow, and ice avalanches) has become an integrated part of natural hazards mapping and mitigation. Starting from the 1960s, depth-averaged models that simulate gravitational mass movements along a prescribed 2D topography were developed (see, e.g., the review by Eglit et al. (2020) for snow avalanche models; Hungr (1995) for debris flows). Later, depth-averaged models have been extended to 3D topography, also aided by higher spatial coverage and resolution of Digital Terrain Models (DTMs). Several depth-averaged models have been proposed to model rock avalanches (e.g., Mergili et al., 2012), debris flows (e.g., Denlinger and Iverson, 2004; McDougall and Hungr, 2004; Iverson and George, 2014; Tayyebi et al., 2021), and snow avalanches (e.g., Christen et al., 2010; Vicari and Issler, 2025). The extensive scientific advancement and adoption of depth-averaged models can largely be attributed to their computational efficiency, as these models modify the 3D conservation equations by integrating them in the vertical direction or perpendicular to the topography—thus reducing the number of conserved flow variables that need to be resolved—and assume simplified rheologies. Additionally, their straightforward integration into Geographic Information Systems (GIS) has also contributed to their popularity, particularly among natural hazard agencies and practitioners focused on hazard mapping.

Recently, advancements in computing power have enabled the resolution of 3D conservation equations within a feasible (yet extensive) timeframe, making it possible to simulate geophysical flows at the slope scale. 3D models have been applied for

debris flows (e.g., Kwan et al., 2015; Koo et al., 2018), snow avalanches (e.g., Gaume et al., 2019; Li et al., 2021; Kyburz et al., 2024), rock and ice avalanches (e.g., Cicoira et al., 2022), and landslides (e.g., Franci et al., 2020; Tran et al., 2024). Nonetheless, although these models are scientifically popular, their use in practical applications for evaluating and mitigating natural hazards remains limited. We believe that this gap can be attributed to multiple factors: (i) inexperience within the natural hazards community and scarcity of guidelines; (ii) limited laboratory experiments to determine the constitutive param-

eters of geomaterials and challenges in determining these parameters in the field; (iii) non-user-friendly software; (iv) higher computational cost compared to depth-averaged models; (v) poor integration of simulation results with GIS tools. The latter issue stems from a mismatch between the output of these 3D *depth-resolved* simulations (i.e., where conservation equations are also resolved in the direction normal to the terrain) and the 2D raster-based formats used in GIS platforms. Indeed, in these simulations, physical quantities such as material positions and their velocities vary continuously along the terrain-normal

direction and are expressed in absolute 3D coordinates. However, 2D raster data can represent only a single value per grid cell, requiring some form of averaging and reduction of the depth-resolved physical quantities to the local topography. This dimensional mismatch limits the straightforward use of 3D simulation results in hazard mapping. Within this work, we aim at addressing this last point by developing a tool to export the results of fully 3D Material Point Method (MPM) simulations into a rasterized format. Specifically, our aim is to convert this depth-resolved MPM particle information into *depth-averaged*

variables that can be saved in a 2D raster format and visualized in GIS. Han et al. (2020) and Su et al. (2024) already proposed algorithms to extract physical quantities from 3D Smoothed-Particle Hydrodynamics (SPH) simulations. Nevertheless, considering the significance of MPM for modeling gravitational mass movements, there is a need for a dedicated algorithm; our method also introduces novel elements, especially concerning the exported depth-averaged variables. We illustrate the practical application of the depth-averaging algorithm by presenting an ice avalanche event simulated using MPM and subsequently

exported to GIS. We then examine the depth-averaged variables derived from the 3D MPM simulation and discuss potential future enhancements of the exporting tool.

## 2   Methods

### 2.1   3D depth-resolved MPM model and material parameters

MPM is a hybrid Eulerian–Lagrangian method commonly used to simulate the behavior of granular and fluid materials within a

continuum mechanics framework. In our case study, we use a Drucker–Prager yield criterion to model the constitutive behavior of ice. However, alternative material models may also be employed within the MPM model to simulate other materials, such as snow or soil (e.g., Gaume et al., 2019; Cicoira et al., 2022). The Drucker–Prager yield function is defined as:

$$q - \gamma(p + a) \leq 0, \tag{1}$$

where $q$ and $p$ are the deviatoric and mean stress, respectively. The material parameters $\gamma$ and $a$ represent friction and at-

traction, respectively. These can be derived from the Mohr-Coulomb yield criterion parameters—i.e., the friction angle $\varphi$ and the cohesion $c$—given that the two yield criteria are assumed to be equivalent under triaxial compression conditions:

$\gamma = (6\sin\varphi)/(3 - \sin\varphi)$ and $a = c/\tan\varphi$. We assume a non-associative flow rule with zero dilatancy. The model incorporates softening, similar to the model of Cicoira et al. (2022), wherein we assume that the friction angle decreases from an initial $\varphi_i = 40°$ to a residual $\varphi_r = 20°$, and cohesion decreases from an initial $c_i = 1$ kPa to 0, depending on the accumulated deviatoric plastic strain $\varepsilon_d^p$. Unlike the simulations of Cicoira et al. (2022), we do not model explicitly erosion of glacier's ice and snow cover. Instead, we assume that the basal friction is higher when the flow is in contact with the glacier ($\mu = 0.65$) than with bedrock or soil ($\mu = 0.55$). This frictional contrast yielded a reasonable match to the observed inundation area. One possible explanation for this assumption is that the erosion of compact, wet snow on the glacier surface may lead to reduction in flow mobility, as also suggested by Li et al. (2022). At $t = 0$, the ice bulk density is set equal to $\rho_0 = \rho_{p0} = 850$ kg/m$^3$, where $\rho_p$ denotes the material points' density. The simulation is carried out with a grid size of $gs = 1$ m, with $n_g = 6$ material points per cell. A complete list of the material parameters is reported in the Supplements (S2).

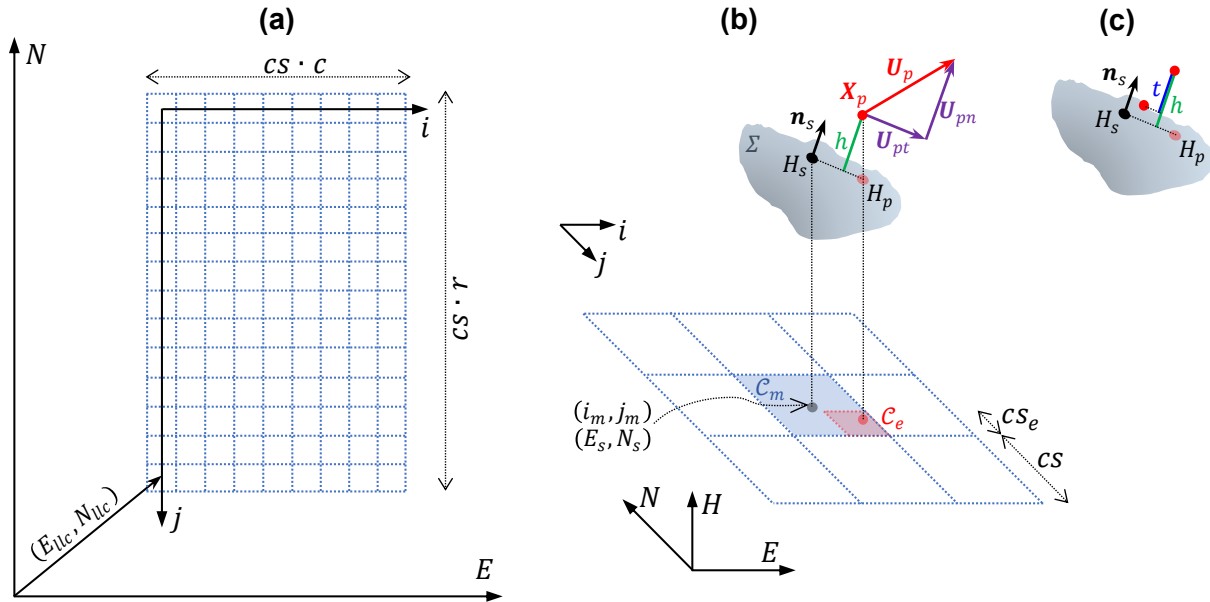

**Figure 1.** Discretization of the topography in ASCII format and calculation of flow-related physical variables: (a) Global and local coordinate system; (b) Calculation of the surface normals and mapping of the material points to their corresponding cell, with definition of flow velocity components, which are parallel ($U_{pt}$) and normal ($U_{pn}$) to the terrain; (c) Schematic of flow depth $h$, defined as the maximum distance from the topography among the material points within a cell; and flow thickness $t$, defined as the difference between the maximum and minimum distances from the topography among all material points within a cell. The two quantities may differ notably when the flow detaches from the terrain.

## 2.2 Topography representation

MPM simulations are performed on a generic topography $\Sigma$, whose elevation is defined in a global Euclidean system ($ENH$) as $H(E, N)$ (Fig. 1). We denote the latitude and longitude coordinates of sampled points on the surface as $(E_s, N_s)$, and $H_s$ their respective elevations on $\Sigma$. Each sampled coordinate $(E_s, N_s)$ specifies the center of a cell with index $(i, j)$, which is

separated by a distance $cs$ (the cell size of the DTM) from the adjacent cell (Fig. 1a):

$$i = \frac{E_s - E_{llc}}{cs}, \tag{2a}$$

$$j = \frac{-N_s + N_{llc} + cs \cdot (r-1)}{cs}, \tag{2b}$$

where $(E_{llc}, N_{llc})$ define the planar coordinates of the centre of the lower left cell of the DTM. The number of cells of the discretized topography in the east and north directions is indicated by $c$ (i.e., columns) and $r$ (i.e., rows), respectively. The parameters $(E_{llc}, N_{llc})$, $(cs \cdot c, cs \cdot r)$, $cs$, define the location, extension and resolution of the DTM, respectively.

To project the simulation results on the topography, it is necessary to define a terrain orientation metric. The unit vector normal to the topography (outward-pointing) is expressed as (see, Fischer et al., 2012):

$$\boldsymbol{n}_s = \frac{(-\partial_E H, -\partial_N H, 1)^T}{\sqrt{1 + (\partial_E H)^2 + (\partial_N H)^2}}. \tag{3}$$

In the exporting code, the spatial gradients of the surface height are approximated using central difference at each grid point:

$$\partial_E H(i,j) \approx \frac{H(i+1,j) - H(i-1,j)}{2 \cdot cs}, \tag{4a}$$

$$\partial_N H(i,j) \approx \frac{H(i,j-1) - H(i,j+1)}{2 \cdot cs}. \tag{4b}$$

## 2.3 Depth-averaging of MPM results

The results of the MPM simulation at a certain time $t_e$ include particles ($p$) with their position $\boldsymbol{X}_p(t_e) = (X_p, Y_p, Z_p)$ and velocity $\boldsymbol{U}_p(t_e) = (U_p, V_p, W_p)$, expressed in the global coordinate system $ENH$. The algorithm initially finds the indexes $(i_m, j_m)$ correspondent to the DTM's cell $\mathcal{C}_m$ to which each particle $p$ belongs (Fig. 1b):

$$i_m(t_e) = \operatorname*{argmin}_{i \in \mathbb{Z} \cap [0, c-1]} \left| \frac{X_p(t_e) - E_{llc} - cs \cdot i}{cs} \right|, \tag{5a}$$

$$j_m(t_e) = \operatorname*{argmin}_{j \in \mathbb{Z} \cap [0, r-1]} \left| \frac{-Y_p(t_e) + N_{llc} + cs \cdot (r-1) - cs \cdot j}{cs} \right|. \tag{5b}$$

Unlike Han et al. (2020) and Su et al. (2024), which project particles along the terrain normal, our approach uses vertical projection to assign particles to raster cells (see discussion in Sec. 2.3.2). The code offers the possibility to export the depth-averaged results to a new digital model with a different cell size ($cs_e$) than the original DTM's cell size ($cs$). The indices $(i_e, j_e)$ for each cell $\mathcal{C}_e$ in the new model are thus computed using equations similar to Eqs. 5a and 5b, but with $cs_e$.

### 2.3.1 Flow heights

Once the DTM's cell $\mathcal{C}_m$ has been identified, the elevation $H_p$ of the surface at the planar coordinate $(X_p, Y_p)$ is calculated imposing the coplanarity condition

$$\boldsymbol{n}_s^T(i_m, j_m) \cdot [X_p(t_e) - E_s(i_m),\ Y_p(t_e) - N_s(j_m),\ H_p(t_e) - H_s(E_s, N_s)]^T = 0. \tag{6}$$

For each non-empty cell $\mathcal{C}_e$, the maximum flow depth (defined as the maximum distance from the topography $\Sigma$ in the surface-normal direction among all the material points within $\mathcal{C}_e$, see Fig. 1c) can be found as

$$h(i_e, j_e, t_e) = \max_{\boldsymbol{X}_p(t_e) \in \mathcal{C}_e} \left[ (Z_p(t_e) - H_p) \cdot n_{sH}(i_m, j_m) \right], \tag{7}$$

where $n_{sH}$ is the component of the normal-to-topography unit vector in the vertical direction.

Similarly, the maximum flow thickness (defined as the difference between the maximum and minimum distances from the topography $\Sigma$ in the surface-normal direction among all material points within $\mathcal{C}_e$, see Fig. 1c) is calculated as

$$t(i_e, j_e, t_e) = h(i_e, j_e, t_e) - \min_{\boldsymbol{X}_p(t_e) \in \mathcal{C}_e} \left[ (Z_p(t_e) - H_p) \cdot n_{sH}(i_m, j_m) \right]. \tag{8}$$

In both Eqs. 7 and 8, we provisionally neglect the increase in flow height associated to the volume of the material points. The code also allows setting a percentile to get the flow depth and thickness from all heights and thickness values of all particles within each cell, thus potentially excluding scattered particles.

### 2.3.2 Flow velocities

The velocity $\boldsymbol{U}_p$ of each particle $p$ within a cell $\mathcal{C}_e$ is decomposed into the slope-normal and slope-parallel components (Fig. 1b):

$$\boldsymbol{U}_{pn}(\boldsymbol{X}_p, t_e) = [\boldsymbol{U}_p(\boldsymbol{X}_p, t_e) \cdot \boldsymbol{n}_s(i_m, j_m)]\, \boldsymbol{n}_s(i_m, j_m), \tag{9}$$

$$\boldsymbol{U}_{pt}(\boldsymbol{X}_p, t_e) = \boldsymbol{U}_p(\boldsymbol{X}_p, t_e) - \boldsymbol{U}_{pn}(\boldsymbol{X}_p, t_e). \tag{10}$$

The slope-normal and slope-parallel velocities can therefore be depth-integrated using (density-weighted) Favre-averaging:

$$\tilde{\boldsymbol{U}}_n(E, N, t_e) = \frac{1}{\int_0^h \rho(E, N, z, t_e)dz} \int_0^h \rho(E, N, z, t_e) \boldsymbol{U}_n(E, N, z, t_e)dz, \tag{11}$$

$$\tilde{\boldsymbol{U}}_t(E, N, t_e) = \frac{1}{\int_0^h \rho(E, N, z, t_e)dz} \int_0^h \rho(E, N, z, t_e) \boldsymbol{U}_t(E, N, z, t_e)dz, \tag{12}$$

where $z$ is the coordinate normal to the topography and $\rho$ is the flow *bulk* density. However, in the code, the integration is not performed in the normal-to-topography direction, but vertically for all particles within a cell $\mathcal{C}_e$. Although this is formally not consistent with the definitions of the depth-averaged velocities of Eqs. 11 and 12, this calculation avoids inconsistencies on concave-up topographies (i.e., duplication of particles to multiple cells is avoided; see also the discussion in Hergarten, 2024, p. 783) and carries substantial simplifications in coding. We therefore calculate the depth-averaged velocities by replacing $dz$ with the discrete particle volume $\mathcal{V}_p$, and $\rho$ with the particle density $\rho_p$:

$$
\begin{aligned}
\tilde{\boldsymbol{U}}_n(i_e, j_e, t_e) &\approx \frac{1}{\sum_{\boldsymbol{X}_p(t_e) \in \mathcal{C}_e} \rho_p(\boldsymbol{X}_p, t_e) \mathcal{V}_p(\boldsymbol{X}_p, t_e)} \sum_{\boldsymbol{X}_p(t_e) \in \mathcal{C}_e} \rho_p(\boldsymbol{X}_p, t_e) \boldsymbol{U}_{pn}(\boldsymbol{X}_p, t_e) \mathcal{V}_p(\boldsymbol{X}_p, t_e) \\
&= \frac{1}{n_p(t_e)} \sum_{\boldsymbol{X}_p(t_e) \in \mathcal{C}_e} \boldsymbol{U}_{pn}(\boldsymbol{X}_p, t_e),
\end{aligned}
\tag{13}
$$

$$
\begin{aligned}
\tilde{\boldsymbol{U}}_t(i_e, j_e, t_e) &\approx \frac{1}{\sum_{\boldsymbol{X}_p(t_e) \in \mathcal{C}_e} \rho_p(\boldsymbol{X}_p, t_e) \mathcal{V}_p(\boldsymbol{X}_p, t_e)} \sum_{\boldsymbol{X}_p(t_e) \in \mathcal{C}_e} \rho_p(\boldsymbol{X}_p, t_e) \boldsymbol{U}_{pt}(\boldsymbol{X}_p, t_e) \mathcal{V}_p(\boldsymbol{X}_p, t_e) \\
&= \frac{1}{n_p(t_e)} \sum_{\boldsymbol{X}_p(t_e) \in \mathcal{C}_e} \boldsymbol{U}_{pt}(\boldsymbol{X}_p, t_e).
\end{aligned}
\tag{14}
$$

Since the (solid) mass of each particle, $\rho_p \mathcal{V}_p$, is constant in MPM[1], the expressions on the far right of Eqs. 13 and 14 are derived, where $n_p$ is the number of particles in each cell $\mathcal{C}_e$. Notably, in this context, the Favre-averaged velocities correspond to the *cell-averaged* velocities. In certain circumstances, however, the flow may dilate, which would lead to a reduction in the flow bulk density $\rho$ from its initial value $\rho_{p0}$—this change in bulk density is not always purely due to plastic volumetric deformation from the constitutive model (i.e., resulting in the particles' density changing from $\rho_{p0}$ to $\rho_p$; see, also, Li et al., 2021), but can also result from the formation of cracks and voids between granules of particles, as well as particles moving apart while air gets mixed into the flow (even though air is not explicitly modeled). Therefore, it would become necessary to replace $\rho_p \mathcal{V}_p$ in Eqs. 13 and 14 with $[\rho \mathcal{V} = \rho_{p0} \mathcal{V}_{p0} + \rho_a (\mathcal{V} - \mathcal{V}_{p0})]$, where $\rho_a$ is the air density and $(\mathcal{V} - \mathcal{V}_{p0})$ is the extra-air volume ingested in the flow. Given that $\rho_a$ is small and assuming limited flow dilation, we neglect the term $\rho_a (\mathcal{V} - \mathcal{V}_{p0})$.

### 2.3.3 Choice of exporting parameters for GIS visualization

GIS results are exported with a specified cell size $cs_e$. Choosing an appropriate value for $cs_e$ is crucial: If the cells are too small, it may result in too many empty cells, especially in areas of the flow where the flow is dilated and particles are dispersed; Conversely, if the cells are too large, it might overestimate the flow extent, especially in areas where the flow has compacted. For an initial estimate, one can use the reference volume $\mathcal{V}$ around a particle as a surrogate for cell size. This volume is derived from mass conservation of a MPM particle, neglecting the extra-air mass (see Sec. 2.3.2), such that $\rho \mathcal{V} \approx \rho_{p0} \mathcal{V}_{p0}$, and therefore:

$$
cs_e \approx \sqrt[3]{\mathcal{V}} \approx \sqrt[3]{\frac{\rho_{p0} \mathcal{V}_{p0}}{\rho}}.
\tag{15}
$$

---

[1]This assumption holds for the temporal evolution of a given particle's mass ($\rho_p \mathcal{V}_p = \rho_{p0} \mathcal{V}_{p0}$) and requires that the densities of various materials within a certain cell $\mathcal{C}_e$ are the same. In this study, we simulate only a single material type (ice), and so the mass of the particles remains unchanged.

As the flow bulk density $\rho$ may not be constant spatially and temporally, the choice of $cs_e$ requires some compromise and may cause the total flow mass (as derived from the exported flow height results) to be not conserved at all time steps. For instance, at the initial time step, results may be exported with $cs_e(t_0) \approx \sqrt[3]{\mathcal{V}_{p0}} = \sqrt[3]{gs^3/n_g}$. If, say, during the flow, $\rho$ becomes smaller than $\rho_{p0}$, results should ideally be exported with $cs_e(t_e) > cs_e(t_0)$. This becomes particularly difficult when results must be exported not just at a specific time step, but also require the export of spatial distribution of the maximum value of a certain depth-averaged variable $\mathcal{F}$ (any of Eqs. 7, 8, 13, 14) over all time steps (this forcedly requires selecting a constant $cs_e$):

$$\mathcal{F}_{\max}(i_e, j_e) = \max_{t_e = k \cdot \Delta t_e, \ k=0,1,\ldots,\frac{t_{\text{sim}}}{\Delta t_e}} \mathcal{F}(i_e, j_e, t_e), \tag{16}$$

where $t_{\text{sim}}$ is the simulated flow duration and $\Delta t_e$ is the exported time step. Furthermore, to avoid spatial-loss of information of maximum values of the depth-averaged flow variables, $\Delta t_e$ should be chosen as

$$\Delta t_e \approx \frac{cs_e}{\|\tilde{\boldsymbol{U}}\|_{\max}}, \tag{17}$$

where $\|\tilde{\boldsymbol{U}}\|_{\max}$ is the maximum (over time and space) depth-averaged speed projected in the $EN$ plane.

## 3   Results and discussion

We first validate the exporting tool on a simplified case of a frictionless block moving on a 2D inclined plane and then falling from a cliff (see Supplements, S1). Subsequently, we simulate in MPM the ice avalanche that resulted from the collapse of a $150000 \text{ m}^3$ portion of the Whymper serac (Courmayeur, Italy) in 1998 (Fig. 2a). The results are exported using $cs_e = 1 \text{ m}$ (estimated from Eq. 15) and $\Delta t_e = 0.2 \text{ s}$ and visualized in Figs. 2b–g. Since the current version of the code is not parallelized, and consequently considerably slow, we provisionally used a large time step $\Delta t_e$, about 10 times greater than what suggested by Eq. 17.

The simulated deposit is shown in Fig. 2b (flow depth $h$, normal to the terrain) and the flow inundation area is shown in Fig. 2c (plotted in terms of maximum depth-averaged slope-parallel flow speed). The runout and flow extent reasonably match the mapped extent of the event. Nonetheless, the simulated left bank lobe of the avalanche stops approximately $300 \text{ m}$ up-slope compared to the mapped runout. Similarly, the right lobe of the simulated avalanche exhibits a shorter runout compared to the actual event. This reduced simulated mobility may result from excessive material deposition in the crevasses (the real terrain DTM at the time of the ice avalanche is unknown and could have had smaller crevasses than the DTM used for simulations) and not explicitly modeling the erodible glacier cover. Furthermore, we only made a few attempts to alter the parameters of the ice material in MPM (in particular, the residual friction and the basal frictions) without further optimization of the simulation results.

We now briefly analyze the influence of complex terrain topography on flow behavior, which can be effectively captured by the depth-averaged results, shown in Figs. 2d–i at $t_e = 50 \text{ s}$. In regions with curved topographies and jumps (e.g., cliffs, crevasses), high values of the depth-averaged slope-normal speed (see Fig. 2d) are calculated, whereas on more gentle terrain the ice flows approximately parallel to the terrain (see Fig. 2e). Specifically, on convex-up topographies along the flow direction

(such as overhangs, or cliff edges), the ice flows in the out-of-slope direction (i.e., positive, blue values of $\boldsymbol{n}_s^T \cdot \tilde{\boldsymbol{U}}_n$). In contrast, on concave-up sections (such as gullies or terrain depressions) the flow is directed toward the topography (i.e., negative, red values of $\boldsymbol{n}_s^T \cdot \tilde{\boldsymbol{U}}_n$). Figure 2h shows the calculated depth-averaged velocities extracted along a linear profile of the glacier (the profile line is shown in Fig. 2d–g). A cliff is located between $\approx 60$ m and $\approx 80$ m of this profile, causing the flow to separate from the topography ($\boldsymbol{n}_s^T \cdot \tilde{\boldsymbol{U}}_n > 0$) at the jump and subsequently collide with the topography ($\boldsymbol{n}_s^T \cdot \tilde{\boldsymbol{U}}_n < 0$) upon landing. In other areas where the curvature of the topography is smaller and there are no terrain jumps, the calculated slope-normal component is smaller, and mostly negligible compared to the slope-parallel velocity component. The flow detachment from the topography is also evident by comparing the flow thickness (Fig. 2f) to the flow depth (Fig. 2g). Figure 2i presents the comparison of these two parameters along the profile line: across the jump and upon landing (splash), the flow depth significantly exceeds the flow thickness—signifying the flow's detachment from the surface (see also Fig. 1c)—whereas the values of $t$ and $h$ are not so far apart when the flow sticks to the terrain (see also the additional analysis in the Supplements, S3). Unlike in depth-averaged models, the detachment from the terrain in our 3D simulations allows to correctly capture that, while in airborne motion, the material does not experience basal flow resistance. Similarly, when the flow impacts the terrain, the 3D MPM can effectively capture the predominant motion towards the terrain, potentially leading to increased basal flow resistance, as well as compaction and/or dilation of the material. This may lead to more precise dynamics compared to depth-averaged models, particularly on complex terrains that exhibit jumps and substantial curvature[2]. These spatial insights highlight how our export tool reveals terrain-driven flow dynamics that are not easily interpreted directly from raw 3D MPM outputs.

---

[2]Depth-averaged models may also somehow mimic reduced/increased basal friction when flowing over convex/concave topographies, if centrifugal curvature-induced effects—thus reducing/increasing the slope normal stress—are being implemented. Moreover, frictional parameters are, in some operational models (e.g., Bartelt et al., 2017), adjusted heuristically based on the curvature of the terrain.

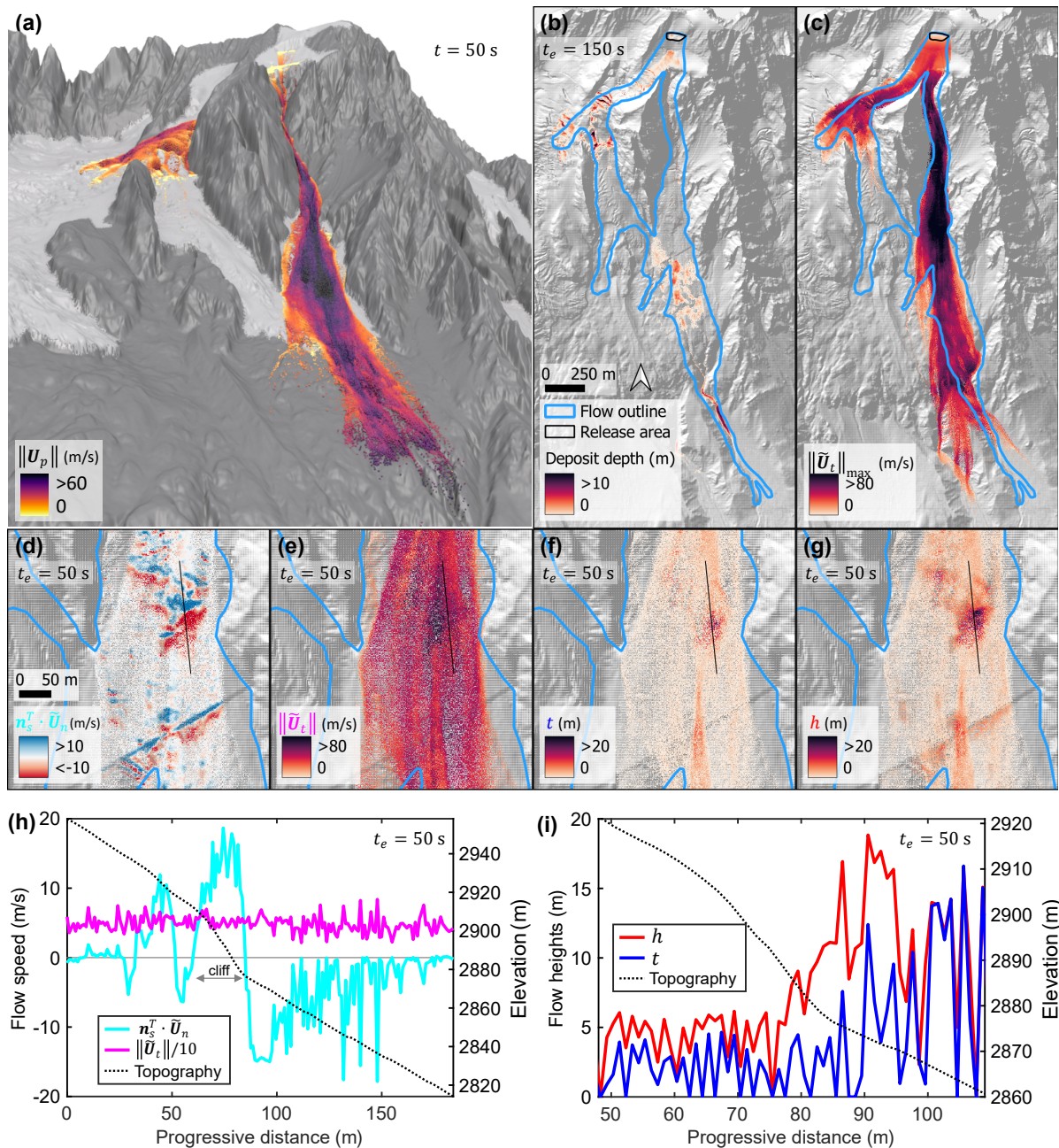

**Figure 2.** Depth-resolved MPM simulations and GIS visualization of exported depth-averaged results. (a) 3D visualization at $t = 50$ s of the MPM simulation of the Whymper ice avalanche in 1998. (b) Exported final deposition depth ($h(t_e = 150$ s)). (c) Maximum depth-averaged slope-parallel speed over all time steps ($\tilde{U}_{t,\max}$). (d) Depth-averaged slope-normal velocity ($\tilde{U}_n$) at $t_e = 50$ s. (e) Depth-averaged slope-parallel speed ($\tilde{U}_t$) at $t_e = 50$ s. (f) Flow thickness ($t$) at $t_e = 50$ s. (g) Flow depth ($h$) at $t_e = 50$ s. (h) Slope-normal ($\tilde{U}_n$) and slope-parallel ($\tilde{U}_t$) speeds at $t_e = 50$ s along a profile line. (i) Flow thickness ($t$) and depth ($h$) at $t_e = 50$ s along a profile line (black line in d–g).

## 4   Conclusion and outlook

In this study, we presented a methodology to transform 3D MPM simulation outputs into 2D rasterized formats that can be visualized with GIS tools. This methodology entails the conversion of *depth-resolved* MPM variables into *depth-averaged* variables. More specifically, we described the computation of flow height metrics, like flow depth and thickness, and flow velocity metrics, like Favre-averaged slope-normal and slope-parallel velocities. Thus, this approach facilitates harvesting the benefits of 3D numerical methods (Li et al., 2021; Cicoira et al., 2022), including the simulation of geophysical flows over complex topographies, terrain jumps, cavities, and interacting with mitigation structures, while enabling practitioners to visualize the model results on GIS maps similar to traditional depth-averaged simulation outputs. Therefore, this tool represents a step forward in using 3D MPM for hazard assessment and mapping of geophysical flows.

The application of the depth-averaging tool to the simulation of an ice avalanche highlighted how complex terrain, like with cliffs and crevasses, can significantly alter flow velocities, causing them to deviate from being parallel to the ground. The computation of slope-normal velocities and the distinction of flow depth (distance between the terrain surface and the highest flow particle) and flow thickness (projected distance between the highest and lowest particles in each cell) revealed that the material might detach from the terrain during jumps and impact the terrain upon landing, potentially influencing flow dynamics through dilation, compaction, and variations in internal and basal stresses. Additional research will be needed to study in detail the effect of complex topographical features on avalanche characteristics. Furthermore, we observed that crevasses along the flow path could lead to significant deposition, further influencing subsequent flow dynamics. Thus, selecting an appropriate DTM appears critical for simulating ice avalanches over glaciers: this needs to be further investigated in the future. Alternative constitutive models for ice, such as incorporating rate dependency, will be required in future studies to achieve better agreement between the simulated ice avalanche and the actual observed runout.

The depth integration of 3D simulation results presented in this work will allow future comparison between 3D MPM simulations and depth-averaged simulations (Wirbel et al., 2024) to verify the assumptions made in depth-averaged numerical models, such as neglecting flow velocities normal to the terrain, assuming the flow sticking to the terrain, constant bulk flow density, and uniform velocity profiles. Additional work is required to broaden the utility of our depth-averaging tool. In the future, we plan to export density and velocity profiles along the flow depth direction. These data could be crucial for determining the pressure distribution of avalanches impacting obstacles. Such *vertical* information (e.g., Kyburz et al., 2024) may, for example, be exported onto raster, to replace traditional empirical analytical models of impact pressure, which (typically) rely solely on depth-averaged density and velocity. The tool could also be used to determine topography changes and erosion rates when an erodible bed layer is explicitly modeled in MPM (e.g., Li et al., 2022).

Finally, a similar framework could be applied in the future to depth-average other 3D particle-based models' results, like SPH and DEM (Discrete Element Method). Recently, similar algorithms were proposed by Han et al. (2020) and Su et al. (2024) to extract depth-averaged physical quantities from 3D SPH simulations. Specifically, the algorithm by Han et al. (2020) assigns each 3D SPH particle to a boundary particle by projecting particle positions along the terrain-normal direction. Physical quantities (e.g., flow depth and basal velocities) are then computed for each boundary particle based on all SPH particles

falling within its associated cell. In contrast, Su et al. (2024) identifies contributing SPH particles as those located within a cylindrical region whose longitudinal axis is centered on each boundary particle and aligned with the terrain normal. Our approach differs in that MPM particles are assigned to 2D raster cells based on vertical projection. While the computation of flow depth and velocity components is conceptually similar across these methods, our implementation includes additional features: the ability to exclude scattered particles from the flow depth calculation, and the explicit computation of the flow thickness. While Han et al. (2020) and Su et al. (2024) assign results to boundary particles, they can also export the data to 2D grids visualized in ParaView. Our method similarly produces rasterized outputs and can directly map quantities to a georeferenced 2D grid, which can be seamlessly exported to a GIS environment. Moreover, in contrast to the mesh-free SPH method, MPM solves momentum conservation on a background Eulerian grid. Thus, an alternative approach could rely on the direct extraction of physical quantities from the MPM code, straightforwardly utilizing the transferred velocities and masses at the grid nodes—thus eliminating the need for a separate export tool.

*Code availability.* The simulations were performed using the MPM code of Gaume et al. (2018). The MPM simulation results were first opened and visualized in the software Houdini. The depth-averaging code was implemented as a Python node within Houdini. In the Supplements, we provide the depth-averaging code.

*Author contributions.* All authors contributed to the conceptualization of this work. MLK initiated the code development, and HV wrote the current code and implemented the depth-averaging algorithm. HV conducted the numerical simulations and generated visual representations of the results. HV prepared the manuscript, with input and contributions from all co-authors.

*Competing interests.* The contact author has declared that none of the authors has any competing interests.

*Acknowledgements.* The Digital Terrain Model and extent of the ice avalanche event were provided by Fondazione Montagna Sicura. We are grateful to Zheng Han and an anonymous referee for reviewing our manuscript and for their insightful comments.

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
