# Peer review of "Brief communication: Depth-averaging of 3D depth-resolved MPM simulation results of geophysical flows for GIS visualization"

_EGUsphere, 2024_

## Author Comment (AC1)

**egusphere-2024-3421 – Reply to Reviewer Comments**

(C and R denote comment and reply, respectively. Changes will be highlighted in red and blue in the manuscript)

**Reviewer 1**

We thank Prof. Dr. Zheng Han for taking the time to read in detail our article. We highly appreciate your comments, which will assist us in improving this manuscript. Below you find the replies to your comments.

**C1:** The title of this brief communication is somewhat confused, since the descriptions of "depth averaging" are unclear to the readers, who cannot to access what the purpose of this study is. In my opinion, the major contribution of this study is to output the 3D-MPM into a 2D format for visualization within a GIS platform.

**R1:** We fully agree that the main contribution of our work is the transformation of 3D MPM simulation results into 2D spatial outputs suitable for visualization and analysis in GIS platforms. However, as shown in the Supplement, the same methodology also applies to "2D" MPM simulations, where momentum conservation is resolved in the direction normal to the terrain, but the flow is constrained within a narrow (plane-strain) strip. To distinguish clearly between 2D/3D MPM simulations and fully depth-averaged approaches, we intentionally use the term "depth-resolved" to refer to MPM simulations that retain the vertical (slope-normal) dimension of momentum conservation. Conversely, we use "depth-averaged" for those simulation results that are integrated in the slope-normal direction to produce 2D (or 1D) fields. Thus, we believe "depth-averaging" appropriately reflects the core operation performed by our tool. Moreover, the terms "depth-resolved" and "depth-averaged" are widely used in the computational geophysics community, and we would prefer to retain this terminology in the manuscript and title for clarity and consistency. See also response **R3**.

**C2:** Page 2, Line 26-30. The authors summarize five bullets of problems in the practical application of 3D models. It is good, however, that bullets from i to iv are not relevant to the topic of this study. I am interested in bullet iv, but it is a pity that the authors did not provide more information on why the integration of 3D simulation results with GIS tools is difficult. I think the more detailed illustration of this issue would be beneficial for the readers to know the significance of this study.

**R2:** We have expanded the discussion of point (v) in the revised manuscript to more clearly explain the challenges associated with integrating depth-resolved simulation results into GIS environments: "The latter issue stems from a mismatch between the output of these 3D *depth-resolved* simulations (i.e., where conservation equations are also resolved in the direction normal to the terrain) and the 2D raster-based formats used in GIS platforms. Indeed, in these simulations, physical quantities such as material positions and their velocities vary continuously along the terrain-normal direction and are expressed in absolute 3D coordinates. However, 2D raster data can represent only a single value per grid cell, requiring some form of averaging and reduction of the depth-resolved physical quantities to the local topography. This dimensional mismatch limits the straightforward use of 3D simulation results in hazard mapping."

**C3:** Page 2, Line 31. The description of "depth-resolved" appears unclear and may even cause confusion that the MPM model in this paper is 2D-based. I suggest replacing it with the more intuitive expression "fully 3D" or "3D" which also aligns better with the subsequent use of "3D MPM" throughout the manuscript.

**R3:** In our work, we use "depth-resolved" specifically to distinguish between simulations that resolve momentum conservation in the direction normal to the terrain (whether in 2D or 3D, see response **R1**), and those that solve and produce depth-averaged outputs. We have now clarified what we mean by "depth-resolved" in the text.

**C4:** Page 2, Line 33. The authors have mentioned that a very similar tool by Su et al. (2024) has been reported. I agree that each model, e.g., MPM and SPH, should require dedicated algorithms; however, the connections and similarities between the methods in the brief communication and the previous paper should be illustrated in the introduction section or better discussed and compared in a separate discussion section.

**R4:** We have now included a discussion in Section 4 about the differences and similarities of our method compared to that of Han et al. (2020) and Su et al. (2024): "Recently, similar algorithms were proposed by Han et al. (2020) and Su et al. (2024) to extract depth-averaged physical quantities from 3D SPH simulations. Specifically, the algorithm by Han et al. (2020) assigns each 3D SPH particle to a boundary particle by projecting particle positions along the terrain-normal direction. Physical quantities (e.g., flow depth and basal velocities) are then computed for each boundary particle based on all SPH particles falling within its associated cell. In contrast, Su et al. (2024) identifies contributing SPH particles as those located within a cylindrical region whose longitudinal axis is centered on each boundary particle and aligned with the terrain normal. Our approach differs in that MPM particles are assigned to 2D raster cells based on vertical projection. While the computation of flow depth and velocity components is conceptually similar across these methods, our implementation includes additional features: the ability to exclude scattered particles from the flow depth calculation, and the explicit computation of the flow thickness. While Han et al. (2020) and Su et al. (2024) assign results to boundary particles, they can also export the data to 2D grids visualized in ParaView. Our method similarly produces rasterized outputs and can directly map quantities to a georeferenced 2D grid, which can be exported in a GIS environment."

**C5:** The title of section 2.2 is too short, making it difficult to know the purpose of this section. It is better to replace it with "topography representation".

**R5:** We have now changed the title of section 2.2.

**C6:** Page 4, Line 75-80. The algorithm to determine the cell to which each particle belongs is quite similar to the previous one that has already been presented in Han et al. (2020). The citation should be added. [Han, Z., Su, B., Li, Y.G., Dou, J., Wang, W.D., Zhao, L.H., 2020. Modeling the progressive entrainment of bed sediment by viscous debris flows using the three-dimensional SC-HBP-SPH method. Water Res. 116031. ttps://doi.org/10.1016/j. watres.2020.116031]

**R6:** We have included the citation to Han et al. (2020), and discussed the similarities in both Sections 2.3 and 4 (see also our response **R4**).

**C7:** Page 4, Line 90. As equation 7 shows, the flow height of the DTM's cell is determined by the largest Z-position of the particles in the cell. Does this dealing way overestimate the flow height? Imagine that 100 particles have been recognized in the cell, 99 have a smaller Z-position, but the last 1 has a greater Z-position. Is it rational to take the largest Z-position for calculating flow height in the cell?

**R7:** As already described in Section 2.3.1, our algorithm allows setting a percentile (P) value to exclude particles scattered at the top of the flow from the calculation of the flow depth and thickness. As a default in the visualization of Figure 2, we used P=100. We have now added an analysis in the Supplements about the influence of different percentile values on the calculated values of the flow heights:

"The depth-averaging algorithm enables us to capture flow detachment from the terrain, as observed in the slope-normal velocity component (Fig. 2h) and the difference between flow depth (measured from the terrain surface to the upper flow surface) and flow thickness (distance between the lowest and highest particles in a cell) (Fig. 2i). Figure 2i was computed using all particles (i.e., percentile set to P=100), including potentially scattered or isolated particles at the top of the flow.

To investigate the influence of filtering these outliers, we compare results obtained using different percentile values (P=20 to P=100), as shown in Fig. R1. Lower percentiles exclude the topmost particles, reducing both flow depth and flow thickness. In general, the trends along the profile remain similar regardless of the chosen percentile. In cells with sparse particle populations, percentile filtering has limited or no effect, as only a few particles are present (see Fig. R2). In zones where the flow is detached from the topography, using lower percentile values significantly reduces the flow thickness $t$, while the flow depth $h$ remains comparatively stable, since it is still measured from the terrain to the upper percentile-bound particle. In contrast, in zones where the flow is attached to the terrain, changing the percentile value affects both flow depth and flow thickness in a similar way."

[Figure]

**Figure R1.** Flow depth (a) and flow thickness (b) exported with different percentile values. The shaded areas show the variability of the flow height metrics for the considered range (between P=20 and P=100).

[Figure]

**Figure R2.** Number of particles per cell at $t_e = 50$ s.

**C8:** Page 4, Line 95-98. I am confused about the physical difference between maximum flow depth and maximum flow thickness in this context. Intuitively, they appear to represent the same quantity. I suggest the authors clearly illustrate the relationship between the two in Figure 1 for better clarity. Is the difference only apparent when particles detach from the ground and scatter into the air?

**R8:** A schematic of the difference between flow depth and flow thickness is being reported in Figure 1c. We have also added a more detailed description in the figure caption: "(c) Schematic of flow depth $h$, defined as the maximum distance from the topography among the material points within a cell; and flow thickness $t$, defined as the difference between the maximum and minimum distances from the topography among all material points within a cell. The two quantities may differ notably when the flow detaches from the terrain."

Furthermore, note that the mathematical definition of the flow thickness was imprecisely defined in the original submission; we have therefore updated the equation as follows:

$$t(i_e, j_e, t_e) = h(i_e, j_e, t_e) - \min_{\boldsymbol{X}_p(t_e) \in \mathcal{C}_e} \left[ (Z_p(t_e) - H_p) \cdot n_{sH}(i_m, j_m) \right]. \tag{R1}$$

The text description of flow depth and flow thickness has also been improved for clarity:

- "[...] the maximum flow depth (defined as the maximum distance from the topography $\Sigma$ in the surface-normal direction among all the material points within $\mathcal{C}_e$, [...]".

- "[...] the maximum flow thickness (defined as the difference between the maximum and minimum distances from the topography $\Sigma$ in the surface-normal direction among all the material points within $\mathcal{C}_e$, [...]".

**C9:** Page 6, Line 130-135. I agree with the authors that cell size is difficult to choose. I am also glad to see that the authors have provided a preliminary criterion for estimating cell size. However, the theoretical basis for this equation is unclear. I would appreciate it if the authors could provide more information on it.

**R9:** We have now provided additional information on the theoretical basis for the equation: "This volume is derived from mass conservation of a MPM particle, neglecting the extra-air mass (see Sec. 2.3.2), such that $\rho\mathcal{V} \approx \rho_{p0}\mathcal{V}_{p0}$, and therefore: [...]"

**C10:** Page 7, Line 160-175. The explanation of the flow behavior is too detailed. I suggest the authors could make it compact and brief, because it is not the key of this brief communication.

**R10:** We appreciate the reviewer's suggestion. However, we believe that the ability of depth-resolved models to simulate flow detachment from the basal terrain is one of the key advantages over depth-averaged models, and it directly motivated the development of this work. For this reason, we prefer to retain the current level of detail in Section 3. In fact, to emphasize the importance of this feature, we propose to add a brief mention of flow detachment to the abstract.

**C11:** Page 9, Line 207-208. As the authors commented, it is right that the algorithm in Su et al. (2024) saves the result on boundary particles. However, it is not precise. As in Han et al. (2020), see the reference in bullet 6 of my comment, the flow depth and basal velocities along both directions are exported in accordance with 2D rasterized grid format. The improvement and major contribution of the authors regard the promotion of this kind of idea to the complex topography and GIS platform.

**R11:** Thank you for the clarification of your method. We have modified the text, see **R4**.

**Reviewer 2**

We thank the anonymous Referee 2 for reading our manuscript in detail and insightful comments which will help us to improve the manuscript.

**C1:** The authors focus their attention on MPM, as it is the method they have at their disposal. This is reasonable. However, the proposed methodology appears to be easily generalisable and potentially applicable to other 3D numerical methods used in geophysical flow simulations. I believe this represents an added value of the work. The authors are encouraged to comment on this, emphasising any specific requirements of MPM for the proposed mapping, and providing more detail on how the present approach differs from the analogous SPH contribution available in the literature

**R1:** We fully agree that the proposed depth-averaging and export methodology can be generalized to other 3D particle-based methods, such as SPH, which is fully Lagrangian. In the current version, we focus on MPM because it is the framework we used to simulate the ice avalanche case study. To address the reviewer's suggestion, we have added more detailed discussion in Section 4 comparing our MPM-based approach with similar SPH-based frameworks proposed by Han et al. (2020) and Su et al. (2024), see our response **R4** to Reviewer 1.

**C2:** A similar observation can be made regarding the constitutive model employed. The proposed methodology appears to go beyond the specific model considered. Why do the authors choose to focus on ice avalanches? Are the details provided in Section 2.1 necessary for the purposes of this article?

**R2:** The proposed exporting algorithm is independent of the specific type of geophysical flow. In this paper, we use an ice avalanche as a case study to demonstrate the methodology in a realistic setting. To clarify this, we have updated Section 2.1 to note that alternative constitutive models may be used depending on the geomaterial of interest: "In our case study, we use a Drucker–Prager yield criterion to model the constitutive behavior of ice. However, alternative material models may also be employed within the MPM model to simulate other materials, such as snow or soil (e.g., Gaume et al., 2019; Cicoira et al., 2022). The Drucker–Prager yield function is defined as: [...]"

**C3:** In the same section, it is stated that MPM is a method used to simulate the behaviour of continuous materials. This statement may be misleading, as MPM, although based on governing equations for continua, is extensively applied to granular materials, which can be viewed as discrete media.

**R3:** We agree that while MPM is based on continuum mechanics, it is widely applied to granular materials, which are often modeled as continua despite their discrete nature. To avoid confusion, we have revised the sentence to: "MPM is a hybrid Eulerian–Lagrangian method commonly used to simulate the behavior of granular and fluid materials within a continuum mechanics framework."

**C4:** Finally, the results presented in Figure 2 are not entirely clear, due to both the limited explanation and the use of colours with poor contrast. I recommend improving the figure's quality and clarity of the explanation.

**R4:** We used a perceptually uniform colormap to visualize flow variables (Figure 2b–g), as it provides consistent contrast across the data range, compared to non-uniform colormaps. Some visual limitations may result from the inherent scattering of material points in the MPM simulation of the ice avalanche. However, the key variables are analyzed in detail in Figure 2h,i, which present clearer 1D profiles extracted along the flow path. We have also clarified the figure by adding relevant variable names to the caption and included additional explanation in the main text to support interpretation.

**References**

Cicoira, A., Blatny, L., Li, X., Trottet, B., and Gaume, J.: Towards a Predictive Multi-Phase Model for Alpine Mass Movements and Process Cascades, Engineering Geology, 310, 106 866, https://doi.org/10.1016/j.enggeo.2022.106866, 2022.

Gaume, J., van Herwijnen, A., Gast, T., Teran, J., and Jiang, C.: Investigating the Release and Flow of Snow Avalanches at the Slope-Scale Using a Unified Model Based on the Material Point Method, Cold Regions Science and Technology, 168, 102 847, https://doi.org/10.1016/j.coldregions.2019.102847, 2019.

Han, Z., Su, B., Li, Y., Dou, J., Wang, W., and Zhao, L.: Modeling the Progressive Entrainment of Bed Sediment by Viscous Debris Flows Using the Three-Dimensional SC-HBP-SPH Method, Water Research, 182, 116 031, https://doi.org/10.1016/j.watres.2020.116031, 2020.

Su, B., Li, Y., Han, Z., Ma, Y., Wang, W., Ruan, B., Guo, W., Xie, W., and Tan, S.: Topography-Based and Vectorized Algorithm for Extracting Physical Quantities in 3D-SPH Form and Its Application in Debris-Flow Entrainment Modeling, Engineering Geology, 340, 107 693, https://doi.org/10.1016/j.enggeo.2024.107693, 2024.